# Increased hepatic interleukin-1, arachidonic acid, and reactive oxygen species mediate the protective potential of peptides shared by gut cysteine peptidases against *Schistosoma mansoni* infection in mice

**Hatem Tallima[1,2]\*, Rashika El Ridi[1]\***

**1** Zoology Department, Faculty of Science, Cairo University, Giza, Egypt, **2** Department of Chemistry, School of Sciences and Engineering, The American University in Cairo, New Cairo, Cairo, Egypt

\* htallima@aucegypt.edu (HT); rashika@sci.cu.edu.eg (RER)

## Abstract

### Background

Multiple antigen peptide (MAP) construct of peptide with high homology to *Schistosoma mansoni* cathepsin B1, MAP-1, and to cathepsins of the L family, MAP-2, consistently induced significant ($P < 0.05$) reduction in challenge *S. mansoni* worm burden. It was, however, necessary to modify the vaccine formula to counteract the MAP impact on the parasite egg counts and vitality, and discover the mechanisms underlying the vaccine protective potential.

### Methodology

Outbred mice were immunized with MAP-2 in combination with alum and/or MAP-1. Challenge infection was performed three weeks (wks) after the second injection. Blood and liver pieces were obtained on an individual mouse basis, 23 days post-infection (PI), a time of *S. mansoni* development and feeding in the liver before mating. Serum samples were examined for the levels of circulating antibodies and cytokines. Liver homogenates were used for assessment of liver cytokines, uric acid, arachidonic acid (ARA), and reactive oxygen species (ROS) content. Parasitological parameters were evaluated 7 wks PI.

### Principal findings

Immunization of outbred mice with MAP-2 in combination with alum and/or MAP-1 elicited highly significant ($P < 0.005$) reduction of around 60% in challenge *S. mansoni* worm burden and no increase in worm eggs' loads or vitality, compared to unimmunized or alum pre-treated control mice. Host memory responses to the immunogens are expected to be expressed in the liver stage when worm feeding and cysteine peptidases release start to be active. Serum antibody and cytokine levels were not significantly different between control and vaccinated mouse groups. Highly significant ($P < 0.05$ - $<0.0001$) increase in liver

**Data Availability Statement:** All data are shown in the manuscript and all further information are available in the Supplementary Materials.

**Funding:** The research work is supported by a grant from the Academy of Scientific Research and Technology Science and Technology Center (STC), National Strategy for Genetic Engineering and Biotechnology, Project Number D 81/2021. The funders had no role in study design, data collection and analysis, decision to publish, or preparation of the manuscript.

**Competing interests:** The authors have declared that no competing interests exist.

interleukin-1, ARA, and ROS content was recorded in MAP-immunized compared to control mice.

## Conclusion/Significance

The findings provided an explanation for the gut cysteine peptidases vaccine-mediated reduction in challenge worm burden and increase in egg counts.

### Author summary

Adjuvant-free cysteine peptidases consistently elicited remarkable reduction in challenge schistosome worm burden in outbred rodents, whether used in an enzymatically active or inactive construct. The findings together suggested that peptide sequences shared by these cysteine peptidases may substitute for the whole molecule and form the basis of a safe, cost-effective, chimeric protein vaccine, easy to manufacture and deliver in countries with limited resources. In support, multiple antigen peptide (MAP) construct of two peptides, MAP-1 and MAP-2, showing high homology to helminth gut cysteine peptidases induced 25%-30% reduction in challenge *Schistosoma mansoni* worm burden in outbred mice. It was, however, necessary to modify the vaccine formula to counteract the effect on the parasite egg counts and vitality. Immunization of mice with MAP-2 in combination with alum and/or MAP-1 elicited reduction of around 60% in challenge *S. mansoni* worm burden and no increase in worm eggs' loads or vitality, compared to unimmunized or alum pre-treated control mice. Considerable increase in liver interleukin-1, arachidonic acid, and reactive oxygen species content in MAP-immunized compared to control mice appeared to elucidate the mechanisms underlying the dual impact of the cysteine peptidase-based schistosomiasis vaccine.

## Introduction

Adjuvant-free, enzymatically active or inactive cysteine peptidases, notably *Schistosoma mansoni* cathepsin B1 (SmCB1), and cathepsin L3 (SmCL3), *Schistosoma haematobium* cathepsin L (ShCL), *Fasciola hepatica* cathepsin L1 (FhCL1) and the prototype, papain consistently elicited highly significant ($P < 0.005$) reduction in *S. mansoni* and *S. haematobium* challenge worm burden in outbred mice and hamsters, respectively [1–10]. These findings indicated that protection might be induced independently of the enzymes proteolytic activity, i.e., host hydrolysis products generated following primary and boost immunization are not essential for worm elimination. The findings together suggested that peptide sequences shared by these cysteine peptidases may substitute for the whole molecule and form the basis of a cost-effective, chimeric protein vaccine, easy to manufacture and deliver in countries with limited resources. The vaccine would be safe, because peptide homology with host corresponding molecules leads to limited antibody generation to the immunogens, but precludes autoimmune responses [1,2]. In support, immunization of outbred mice with adjuvant-free, cysteine peptidases-derived MAP constructs, (MAP-1 and MAP-2) elicited modest, type 2-skewed antibody and cytokine responses conducive to significant ($P < 0.05$) reduction of about 30% in challenge *S. mansoni* worm burden. Yet, MAP-2 immunization was associated with egg loads in liver and small intestine not different from infected control mice. The small intestine circumoval granulomas number and diameter were larger than in MAP-1-immunized mice that showed

considerable egg counts in liver and intestine. The mechanism(s) underlying these effects were not explored, but it was necessary to modify the vaccine formulation in an attempt to skew the host responses towards restricting parasite eggs production and vitality. It was recommended to intensify the immune responses to MAP-2, in view of controlling the vitality of the eggs produced by the worms, and only use MAP-1 combined with MAP-2 in order to elicit host responses restricting parasite fecundity [11]. Experiments were, therefore, performed using MAP-1 and MAP-2, alone or in mixture, or combined with alum adjuvant [12,13]. The aims were to increase the challenge worm reduction level, control the caveats of peptide immunization on the egg counts and granulomas formation, and find clues to the mechanism(s) allowing host responses to simple peptides to significantly interfere with challenge worm survival and reproduction.

## Materials and methods

### Ethics statement

All experiments involving animals were conducted according to the ethical policies and procedures approved by the Ethics Committee of the Faculty of Science, Cairo University, Egypt (Approval no. CU/I/F/65/19).

### Multiple antigen peptide synthesis

Peptides IRDQSRCGSSWAFGAVEAMS, and EQQLVDCSYKYGNDGCQGG, showing highest sharing of amino acid sequences with helminth and murine cathepsins B and L, papain, and major allergens were synthesized as endotoxin-free, tetra-branched multiple antigen peptide (MAP) constructs at Thermo Fisher scientific (Waltham, MA, USA), and designated as MAP-1 and MAP-2, respectively [11].

### Mice and parasites

Outbred, female, six week-old CD1 mice were obtained from the Schistosome Biological Supply Program (SBSP) at Theodore Bilharz Research Institute (TBRI) Giza, Egypt and maintained throughout experimentation at the animal facility of the Zoology Department, Faculty of Science, Cairo University. Cercariae of an Egyptian strain of *S. mansoni* were obtained from SBSP/TBRI, and used for infection immediately after shedding from *Biomphalaria alexandrina* snails.

### Experimental plan

Two independent experiments were performed in parallel, and in each, 5 of 45 mice were retained without immunization or schistosome infection, and considered as naïve, while 40 mice were randomly distributed into three groups and vaccinated intramuscularly, twice with a three weeks (wks)-interval. In Experiment 1, each of 13 or 14 mice were injected with immunogen- and adjuvant-free Dulbecco's phosphate-buffered saline, pH 7.1 (D-PBS), 25 μg MAP-2 adsorbed on 130 μg alum (Alhydrogel, Aluminum Hydroxide Gel 13 mg/mL, Sigma-Aldrich-Merck, Darmstadt, Germany), or 12.5 μg MAP-1 +12.5 μg MAP-2 in 100 μL D-PBS. In Experiment 2, each of 13 or 14 mice were immunized with immunogen- and alum-free D-PBS (control mice), immunogen-free alum (130 μg/mouse; adjuvant controls), or 15 μg MAP-1 + 10 μg MAP-2 + 130 μg alum adjuvant. Challenge infection was performed three wks after the second injection via percutaneous exposure of each mouse to 100 viable cercariae of *S. mansoni*, as described [11]. Blood and liver pieces were obtained from 6 to 8 mice per group, 23 days post-infection (PI), a time of schistosome development and feeding in the liver

before mating [14], and immediately processed before storing at -20˚C until use. Serum samples were examined for the levels of circulating antibodies and cytokines. Liver cell extracts were used for assessment of hepatic cytokines, uric acid, arachidonic acid (ARA), and reactive oxygen species (ROS) content. Parasitological parameters were evaluated in 5 to 7 mice per group 7 wks PI. No attempt was made to assess cytokine or antibody responses to the peptide immunogens at this interval because of the confounding strong reactivities to the parasite egg antigens [1–9,11].

## Parasitological parameters

Worm burden was evaluated by hepatic portal venous system and mesenteric blood vessels perfusion as described in detail previously [11]. After perfusion, the liver and small intestine of each mouse were harvested, and 50 mg pieces processed for histological examination. Parasite egg burden of individual mice was evaluated in 200 mg liver or small intestine following incubation in 4% KOH for 1 h at 40˚C as described [1–11]. Percent change in worm and egg burden was evaluated by the formula: % change = [mean number in infected controls − mean number in immunized infected mice / mean number in infected controls] × 100.

Liver and small intestine sections from each mouse were stained with haematoxylin and eosin and examined for the number/field and diameter of circumoval granulomas [6–9]. Hepatic egg granulomas numbers are mean ± SD /field of 5–10 fields per section of 5 mice per group. Granuloma diameters (μm) are shown as mean ± SD of all circumoval granulomas in sections of five mice per group. Photographs were acquired by light microscopy (Olympus, Tokyo, Japan).

## Serum cytokine and antibody assays

Quantitative determination of mouse interleukin (IL)-4, IL-5, and IL-17, and interferon-gamma (IFN-γ) (ELISA MAX Set, BioLegend, San Diego, CA, USA) was evaluated in individual mouse sera using capture enzyme-linked immunosorbent assay (ELISA), following the manufacturer's instructions. The antibody isotypes to a mother cysteine peptidase molecule, recombinant FhCL1 [3], a gift of Professor Dr. John P. Dalton, were determined by indirect ELISA in 1:100-diluted sera, assayed on an individual mouse basis, as described [11]. Absorbances of duplicate wells were evaluated spectroscopically at 405 nm (Multiskan EX, Labsystems, Helsinki, Finland).

## Liver cytokines, uric acid, arachidonic acid, and ROS assays

**Liver extracts preparation and protein content.** The weight of a liver piece from each mouse was recorded before homogenization in D-PBS supplemented with 0.1% Triton X-100, and protease inhibitors: leupeptin (4 μg/mL) and 1 mM phenyl methyl sulfonyl fluoride (Merck). The homogenates were incubated on ice with shaking for 30 min, and then centrifuged at 400 x $g$ for 10 min [15]. The supernatants were retrieved in ice-cold reaction tubes. Liver extracts were assessed for protein content spectrophotometrically at 280 and 260 nm, using the formula: protein concentration mg/mL = 1.55 x $A_{280}$−0.76 x $A_{260}$ and at 595 nm for the Bio-Rad Protein Assay, and stored at -20˚C until use.

**Cytokines analysis.** Quantitative determination of mouse thymic stromal lymphopoietin (TSLP), IL-25, IL-33, IL-1β, IL-10, IL-13 (R&D Systems, Minneapolis, MN, USA), IL-4, IL-5, IL-17, and IFN-γ (BioLegend), was evaluated in individual mouse liver Triton X-100 extracts (200 μg protein per each of duplicate wells) using capture ELISA, following the manufacturer's instructions.

**Uric acid assays.**   Liver Triton X-100 extracts were assayed for uric acid content in duplicate 50 μg protein samples per well using in parallel Uric Acid Assay Kit (ab65344, Abcam, Cambridge, UK), and Uric Acid Kit (Chronolab Systems, S.L., Barcelona, Spain), following the manufacturers recommendations and procedures. Additionally, 10 mg liver were thoroughly homogenized and added with 200 μL uric acid assay buffer (100 mM Tris-HCl, pH 7.5), incubated for 30 min on ice and centrifuged at 5,000 x $g$ for 2 min. The supernatant was retrieved in ice-cold reaction tubes and 2.5 and 5 μL/well in duplicates were immediately examined for uric acid content using the Uric Acid Kit (Chronolab).

**Arachidonic acid assays.**   Free ARA content in liver Triton X-100 extracts [16] was evaluated by capture ELISA. Wells were coated with 250 ng unlabelled rabbit polyclonal antibody to ARA (MyBioSource, San Diego, CA, USA, MBS2003715) overnight at 4˚C. Following washing in 0.1 M PBS/0.05% Tween 20 (PBS-T), 200 μg liver protein of naive, control and immunized mice were added in duplicate wells to a total volume of 100 μL PBS-T, and incubated for 2 h at room temperature. The wells were thoroughly washed and added with 150 ng horseradish peroxidase-linked polyclonal antibody to ARA (MyBioSource, MBS2051576) for 1 h at room temperature. The reaction was visualized 30 min after adding 3,3',5,5' tetramethylbenzidine substrate (Sigma).

Arachidonic acid content was additionally evaluated by immunohistochemistry as described previously [8,9,17], except that liver sections were exposed to 3% hydrogen peroxide (Sigma) to block endogenous peroxidase activity, then incubated with 0 or 0.5 μg horseradish peroxidase-linked polyclonal antibody to ARA (MyBioSource, MBS2051576) overnight at 10˚C. The reaction was visualized with Dako Liquid DAB + Substrate Chromogen System (Agilent Dako, Santa Clara, CA, USA). Photographs were acquired by light microscopy.

**Reactive oxygen species assays.**   2',7'-dichlorodihydrofluorescein diacetate (DCHF-DA) is a cell-permeable non-fluorescent probe. After mixing with cell homogenates, DCHF-DA is deacetylated by cellular esterases to a non-fluorescent compound which is readily oxidized by ROS into a highly fluorescent compound, 2', 7'–dichlorofluorescein [18]. Duplicates of 25, 50, 100 and 200 μg liver proteins of individual mice were incubated with 20 μM DCHF-DA (Merck, D6883) at room temperature, in the dark, for 1 h and ROS release estimated by fluorescence spectroscopy with maximum excitation (Ex) and emission (Em) spectra of 485 nm and 535 nm, respectively (Victor X4 Multi-Label Plate Reader, PerkinElmer, Waltham, MA, USA).

## Statistical analysis

All values were tested for normality. Students'–$t$- 2-tailed, Mann-Whitney, and one-way ANOVA with post test were used to analyze the statistical significance of differences between selected values, and considered significant at $P < 0.05$ (GraphPad InStat, San Diego, CA, USA).

## Results

### Parasitological parameters

Following our recommendation to improve MAP-1 and MAP-2 protective potential [11], MAP-2 emulsified in alum, or combined with MAP-1 was used to immunize mice against challenge *S. mansoni* infection. Highly significant ($P < 0.005$, Mann-Whitney) reduction in total, male and female worm burden was recorded, varying between 52.1 and 61.6% (Table 1). Whilst MAP-2—alum immunization led to substantial, albeit insignificant, decrease in egg load in liver and small intestine, adding MAP-2 to MAP-1 failed to modify the ability of the latter to elicit considerable increase in number of eggs retrieved in small intestine (Table 1).

**Table 1. Effects of MAP-2 + alum or combined with MAP-1 on challenge worm parameters*.**

| | D-PBS | Mice injected with MAP-2 + alum | MAP-1+MAP-2 |
|---|---|---|---|
| **Parameter** | | | |
| **Total worm burden** | | | |
| Mean ± SD | 17.0 ± 2.0 | 7.0 ± 2.5 | 7.4 ± 2.3 |
| P versus infected controls | | 0.0043 | 0.0043 |
| Percent reduction | | 58.8 | 57.6 |
| **Male worm burden** | | | |
| Mean ± SD | 7.3 ± 1.5 | 3.2 ± 1.4 | 2.8 ± 1.7 |
| P versus infected controls | | < 0.005 | < 0.005 |
| Percent reduction | | 56.1 | 61.6 |
| **Female worm burden** | | | |
| Mean ± SD | 9.6 ± 1.5 | 3.8 ± 1.1 | 4.6 ± 1.3 |
| P versus infected controls | | < 0.005 | < 0.005 |
| Percent reduction | | 60.4 | 52.1 |
| **Liver egg counts** | | | |
| Mean ± SD | 13500 ± 8455 | 6400 ± 2607 | 15800 ± 4919 |
| P versus controls | | NS | NS |
| **Small intestine egg counts** | | | |
| Mean ± SD | 16000 ± 9838 | 10200 ± 3701 | 41400 ± 11415 |
| P increase versus controls NS | | | 0.0105 |
| P increase versus MAP-2 + alum | | | 0.0079 |
| **Liver egg granulomas number/field** | | | |
| Mean ± SD | 3.7 ± 0.6 | 3.9 ± 0.3 | 4.0 ± 0.5 |
| P (ANOVA) | | NS | NS |
| **Liver circumoval granulomas diameter (µm)** | | | |
| Mean ± SD | 330 ± 46 | 392 ± 32 | 430 ± 74 |
| P (ANOVA) | | NS | NS |

* Parasitological parameters were evaluated 7 wks post challenge infection. Liver egg granulomas were examined in 5 to 10 fields (10 x 10) in two sections for each of 5 mice per group. P values indicate levels of statistical (ANOVA and Mann-Whitney) differences between immunized and infected control mice. NS = not significant.

Thus, MAP-1 + MAP-2 immunization led to increase in fecundity of surviving worms (Fig 1A and 1B). Despite the differences in worm survival and egg loads, the mean number /field (10 x 10) and diameter of liver (Table 1) and small intestine egg granulomas did not differ between the three groups, as assessed by one-way ANOVA. The data confirm that host responses to MAP-1 are associated with increase in number of liver and small intestine eggs that are characterized by limited viability and immunogenicity [11].

Since it was not recommended to use MAP-2 immunogen without adjuvant, MAP-1 and MAP-2 immunogen mixture was combined with alum, and impact on challenge worm parameters compared to untreated and alum-administered controls. Alum administration failed to affect challenge worm burden or egg counts and granulomas number compared to untreated controls (Figs 2 and 3). MAP-1+ MAP-2+ alum immunization elicited a record (P = 0.0011, Mann-Whitney) reduction in total, male and female worm burden of 65.8%, higher than for any full-length cysteine peptidase tested, including papain [1–11] (Fig 2). The egg counts in liver and small intestine were not increased and did not differ from infected and alum controls (Fig 2), for the first time with MAP-1 inclusion. The number and diameter of liver egg granulomas were similar in the control (untreated and alum-administered) and immunized mice

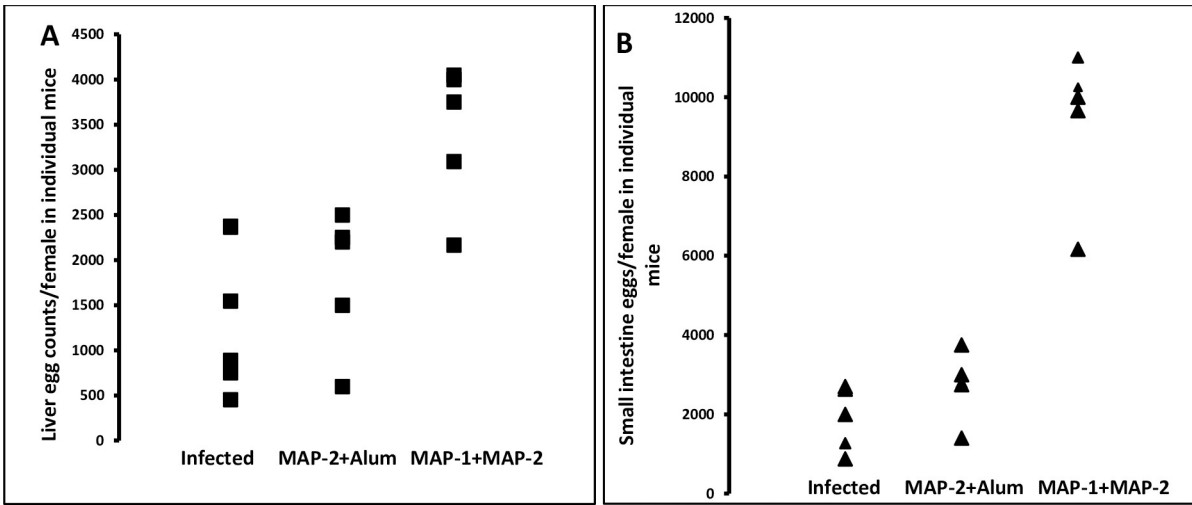

**Fig 1. Parasite egg parameters.** Worm fecundity was evaluated based on liver (A) and small intestine (B) parasite egg counts in individual mice, at wk 7 post infection (PI).

(S1 Fig). Yet, and even more importantly than the reduction in worm burden, immunization with MAP-1 + MAP-2 + alum was associated with significant decrease in number ($P = 0.0378$) and diameter ($P = 0.0003$) of small intestine egg granulomas compared to the control mice (Table 2 and Fig 3). The data indicated that addition of alum or MAP-1 to MAP-2 was associated with impaired parasite egg ability to transit to exit points.

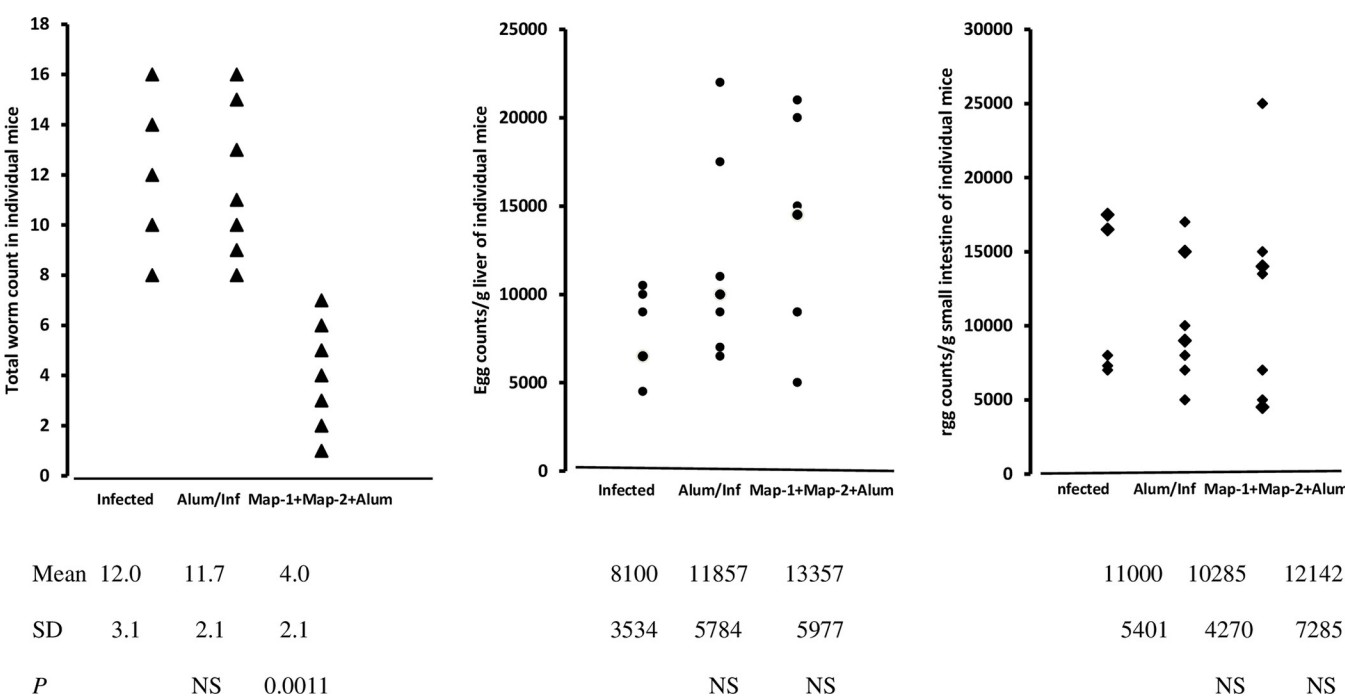

**Fig 2. Parasite worm and egg counts.** Five to 7 mice per group were examined for parasitological parameters 7 wks post challenge infection. *P* values indicate levels of statistical (Mann-Whitney) differences between immunized and infected control mice.

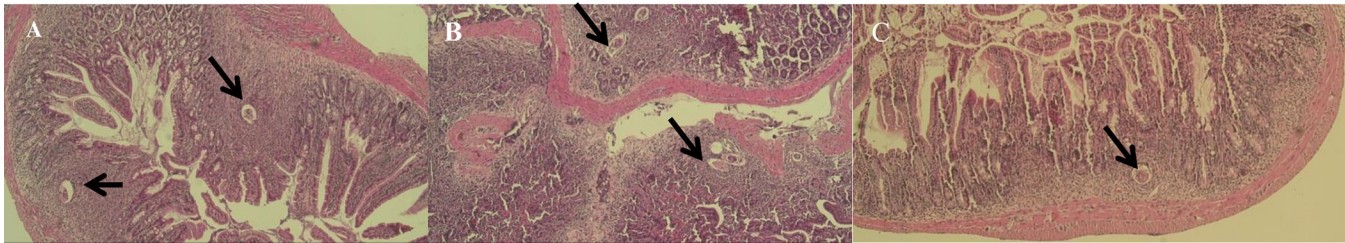

**Fig 3. Small intestine haematoxylin/eosin-stained sections.** D-PBS (A), alum (B), MAP-1+ MAP-2+ alum-administered mice, 7 wks post challenge infection with *S. mansoni*. Typical of 5 mice per group. The arrows point to the egg granulomas. x 100.

## Immunological parameters

**Serum immune responses.** Enough serum was collected from 4 mice and tested on an individual mouse basis for levels of circulating IL-4, IL-5, IL-17 and IFN-γ, and antibody isotype response to a mother cysteine peptidase molecule, on day 23 PI, at time developing worms are still in the liver [14]. Infection with *S. mansoni* in untreated and alum-administered mice elicited increase in serum IL-4, IL-17, and IFN-γ, compared to naïve mice. Serum cytokine levels in MAP-vaccinated mice were lower than in naïve and control infected mice, except for IL-4 and IL-17 ($P < 0.05$) in mice immunized with MAP-1 + MAP-2 (Fig 4A–4D). Serum anti-cysteine peptidase antibodies levels and isotypes were not different among naïve, control and vaccinated mice, except for increase in IgG2a antibodies in MAP-1 + MAP-2 + alum-immunized mice (Fig 4E).

**Liver cytokines.** Liver cells of healthy, untreated and uninfected naïve mice released a plethora of type 1, type 2, and type 17 cytokines. Developing, 23 days-old *S. mansoni* worms released molecules that were not able to modulate the levels of the type 2 cytokines, TSLP, IL-25, IL-4, IL-5, and IL-13, and elicited significant decrease of IL-33 ($P < 0.01$). No impact was recorded on the levels of released IL-1, IL-10, and IFN-γ compared to naïve mice, while the most remarkable change concerned significant ($P < 0.01$) increase in hepatic cell release of IL-17 (Fig 5). Booster alum administration 3 wks before percutaneous infection with *S. mansoni* cercariae was associated liver cells production of significantly ($P < 0.05$- $P < 0.005$) less of each cytokine tested, realizing again balance of cytokine types, only at a lower quantitative

**Table 2. Effects of MAP-1+ MAP-2 + alum on challenge worm egg granulomas in small intestine\*.**

|  | D-PBS | Mice injected with D-PBS + alum | MAP-1+MAP-2+alum |
|---|---|---|---|
| **Small intestine egg granulomas** |  |  |  |
| **Mean number per section** | 2.0 | 2.4 | 1.0 |
| SD | 0.7 | 1.4 | 0.7 |
| *P* one-way Anova |  | 0.0275 |  |
| *P* versus infected controls |  | 0.0491 | 0.0378 |
| **Mean diameter (µm)** | 400 | 360 | 163 |
| SD | 68 | 166 | 80 |
| *P* one-way Anova |  | 0.0064 |  |
| *P* versus infected controls |  | NS | 0.0003 |

\* Small intestine parasite egg granulomas number and diameter (µm) were evaluated in 2 sections of each of 5 mice per group, 7 wks post challenge infection. *P* values indicate levels of statistical differences between immunized and infected control mice.

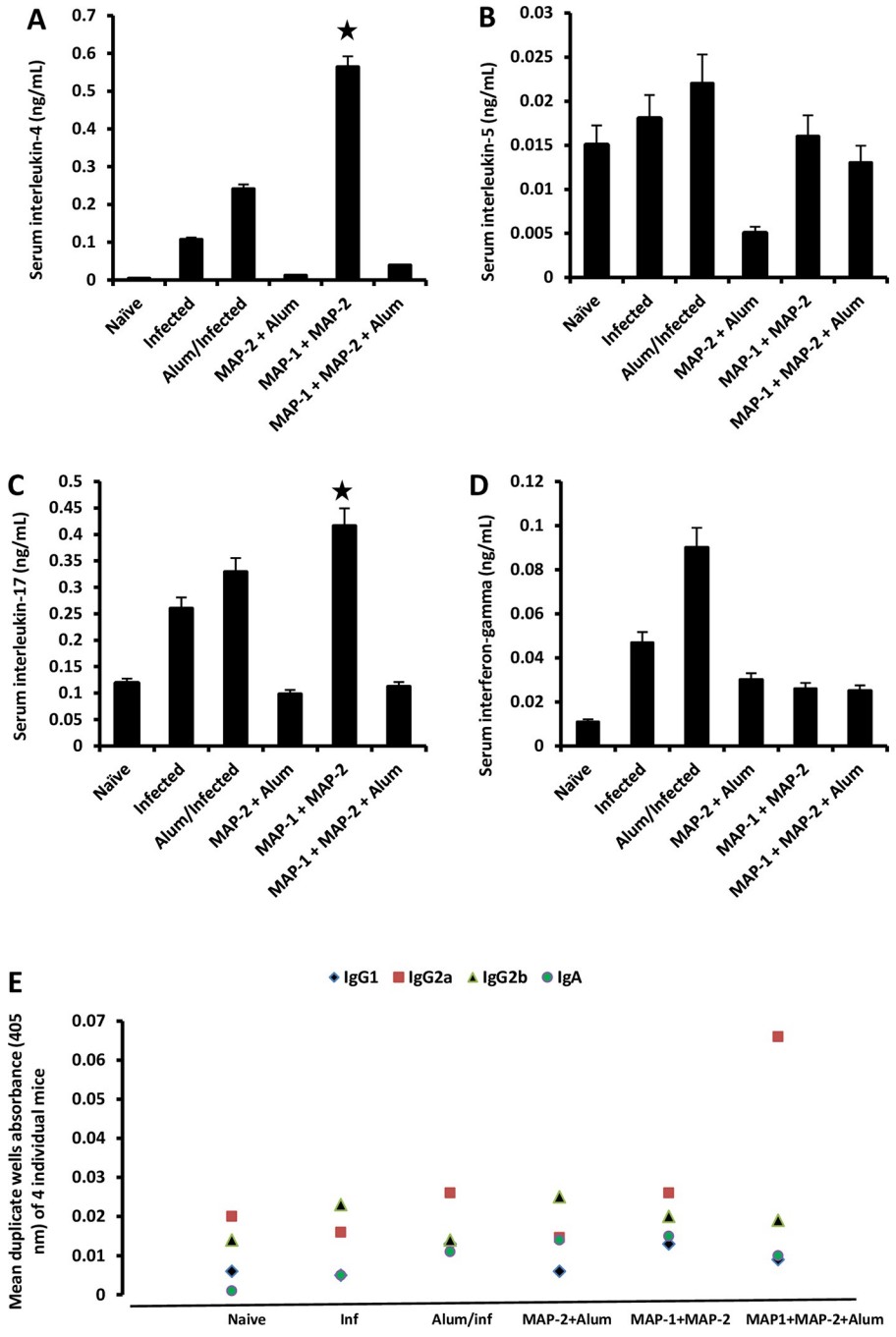

**Fig 4. Serum cytokine and antibody isotypes levels in mice examined 23 days PI.** (A-D), Each column represents mean cytokine levels of 4 mice assessed on an individual basis, and vertical bars the standard error (SE) about the mean. (E), each symbol represents mean absorbance of 4 mice assessed on an individual basis with SE < 5%. Statistical differences were assessed between MAP-vaccinated mice versus naïve and infected controls (Inf). * $P < 0.05$.

level than in naïve animals. Except for IFN-γ, levels of all cytokine tested were significantly ($P < 0.05$- $P < 0.005$) lower when compared to infected mice (Fig 5).

Immunization of mice with MAP-2 + alum was associated with liver cells production of cytokine levels significantly ($P < 0.05$) lower than naïve and infected mice, likely because of

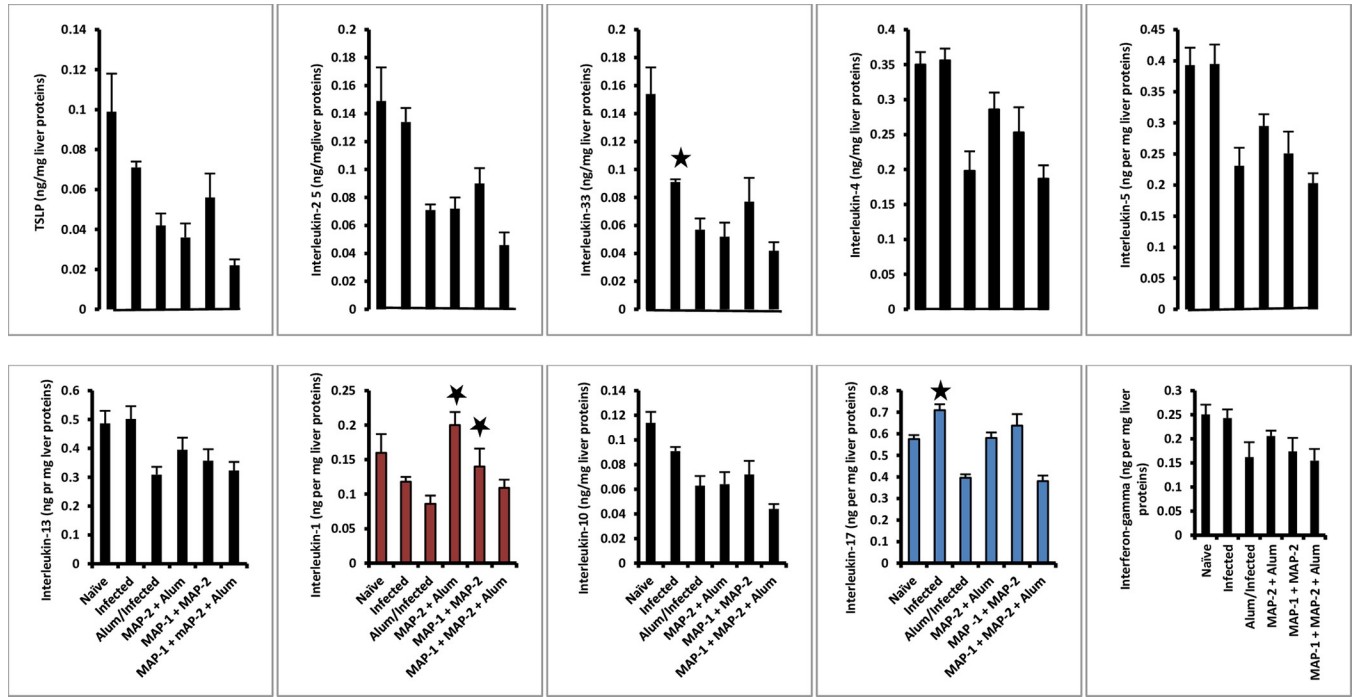

**Fig 5. Cytokine levels in liver Triton X-100 extracts on day 23 PI.** Each column represents ng cytokine/mg liver proteins of 5–8 individual mice/group, and vertical bards denote the SE about the mean. Asterisks denote significant (*P* < 0.05- < 0.005) differences between infected and naïve or MAP-immunized and infected mice.

the impact of alum. Conversely, IL-1 levels were significantly (*P* < 0.005) higher than untreated and alum-administered infected mice. Immunization with alum-free MAP-1 and MAP-2 led to highly significant decrease in challenge worm burden that correlated again with significant (*P* < 0.05) increase in hepatic cells IL-1. Decrease (*P* < 0.05) in liver cells type 1 and type 2 cytokines production, compared to unimmunized infected mice, allowed IL-1 preponderance, perhaps giving a clue for the large increase in production of eggs with limited immunogenicity and viability in this group. MAP-2+MAP-1+ alum immunization induced highly significant (*P* < 0.005) decrease in challenge worm recovery and small intestine pathology, and the most extremely significant decrease (*P* < 0.005) in cytokines tested, namely all type 2 cytokines, IL-10, and IL-17, with the exception of IL-1, which levels did not differ from unimmunized infected mice (Fig 5).

## Biochemical parameters

**Uric acid.**    Liver Triton X-100 extracts were assayed in duplicate wells for uric acid levels using two separate assays. Similar results were obtained and therefore pooled. Twenty three days *S. mansoni* infection whether preceded or not by alum treatment or MAP immunization elicited no statistically (Anova and Mann Whitney tests) significant changes in host liver uric acid content (S2A Fig). Similar results were obtained using liver uric acid buffer extracts (S2B Fig).

**Arachidonic acid.**    Repeat capture ELISA tests confirmed that the levels of liver ARA readily extracted by Triton X-100 [16] significantly (*P* = 0.0007) increased at 23 days PI compared to naïve mice, provided that alum was not administered before infection (Fig 6). Immunization with alum adjuvanted MAP-2 and MAPs mixture overcame (*P* < 0.05) the alum impact

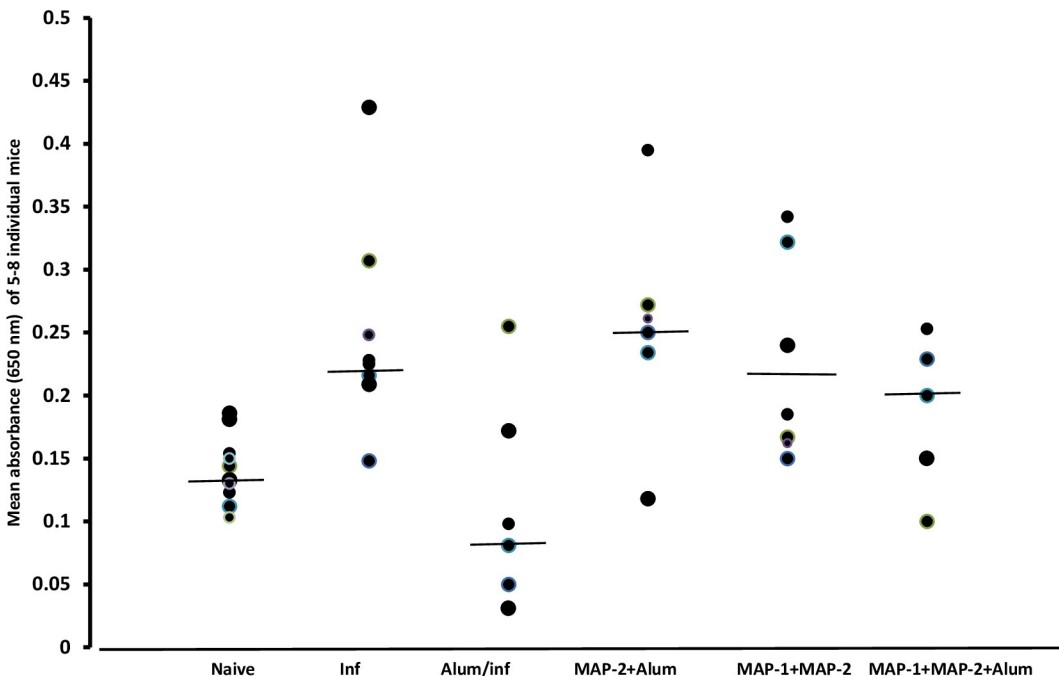

**Fig 6. Liver free arachidonic acid reactivity in capture ELISA.** Each point represents mean of duplicate wells for 5–7 individual naïve, infected (Inf) and MAP-immunized mice, 23 days PI, and horizontal lines depict the median. Values in the different groups significantly ($P = 0.0014$) differed as assessed by ANOVA. Values of mice immunized with MAP-2 + alum, MAP-1 + MAP-2, and MAP-1 + MAP-2+ alum differed significantly from naïve ($P < 0.05 - < 0.005$) and from alum/infected ($P < 0.05$) mice.

and was associated with ARA content significantly higher ($P < 0.05 - < 0.002$) than naïve, but not control infected, mice (Fig 6). Histochemical findings mirrored the capture ELISA results (S3 Fig).

**Reactive oxygen species.**   Results illustrated in Fig 7 and Table 3 show the effect of MAP immunogen formulations on liver ROS content on day 23 PI.

## Discussion

The results obtained in the present study indicated that our recommendations regarding the use of MAP-1 and MAP-2 were judicious as the novel formulations elicited highly significant ($P < 0.005$) of $> 60\%$ to up to 68% (with MAP-1 + MAP-2 + alum) challenge worm burden reduction. Decrease in liver and intestine worm egg counts, immunogenicity, and viability was also achieved.

Attempts at deciphering the mechanism(s) underlying protection were made at the liver stage, 23 days PI, prior to worm maturity, mating, and migration to the final abode in the capillaries of intestine mesenteries and egg laying for two reasons. First, in the liver, developing worms voraciously feed on host erythrocytes, and increasingly produce the cysteine peptidases necessary for digestion [14,19–21]. Release of cathepsins B and L would activate immune memory responses to the MAP immunogens. Second, like for the lung capillaries, the liver sinusoids represent a danger for the migrating worms because of the ease of extravasation to certain demise [2,14,22–24].

Despite that at this time worms are feeding and regurgitate cathepsins B and L-rich products [20], the induced primary (in untreated and alum-administered infected mice) and memory (mice immunized with cathepsins-related MAP) antibody responses to the target

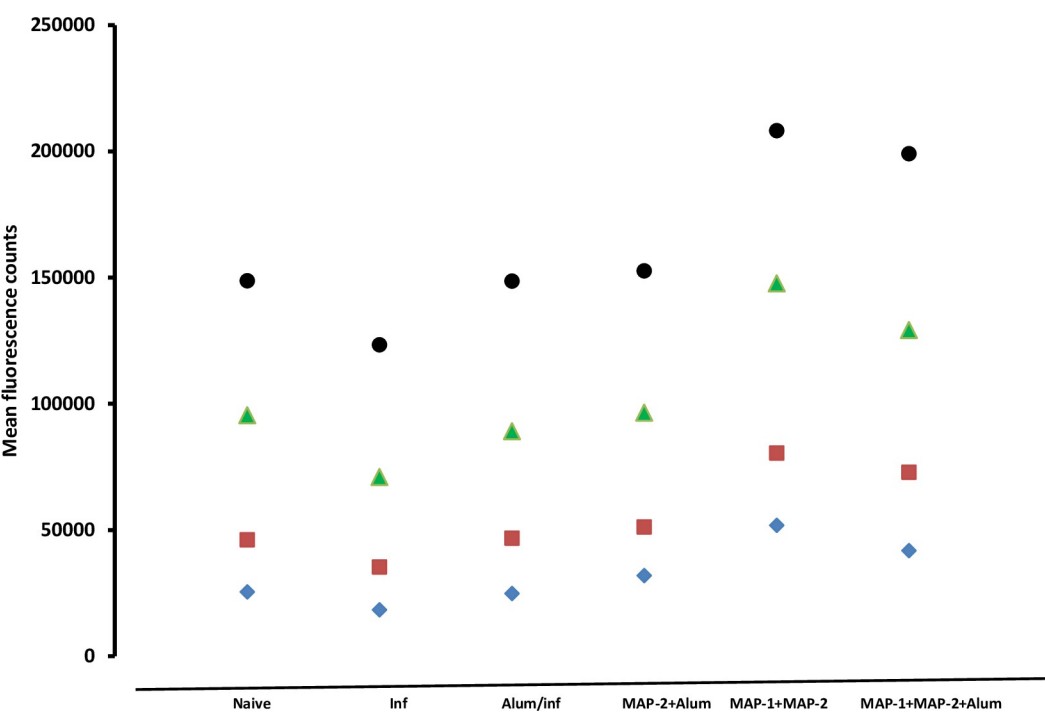

**Fig 7. Reactive oxygen species fluorescence.** Each point represents mean reactivity of 25 (blue diamond), 50 (brown squares), 100 (green triangles), and 200 (black circles) μg liver proteins of 6 to 8 mice per group.

cathepsin molecule were negligible. These data are in full accord with findings in mice reported and discussed previously [11], and in rabbits immunized thrice with SmCB1-derived peptides suspended in complete and incomplete Freund's adjuvant [25]. Alum was not co-administered with the invading schistosome cercariae and was not expected to potentiate the humoral responses to worm-released molecules. However, increase in serum IgG2a antibodies to FhCL1 in MAP-1+MAP-2+alum-immunized mice could be related to alum co-administration [12,13,26]. Mice of this group showed the highest reduction in challenge worm burden and small intestine egg granulomas number and diameter (Table 2 and Figs 2 and 3), likely because of activation of inflammatory immune cells by antibody/released gut cathepsin complexes, a mechanism of worm attrition discussed previously [1,2,9] (Fig 8).

Increase in serum IL-4, IL-17, and IFN-γ levels was noticed in infected control mice, associated with developing worms secretion of SmCB1 [27]. Alum co-administered with the MAP immunogens failed to potentiate their serum cytokine response. Indeed, MAP immunization was associated with serum cytokine levels lower than in infected control mice, except for increase in IL-4 and IL-17 in adjuvant-free MAP-1 + MAP-2-immunized mice, showing the only cases of large egg loads in liver and small intestine.

Of interest, the hepatic levels of cytokines, ARA, and ROS significantly ($P < 0.05$ - $<0.0001$) differed among the various mouse groups. The balance of type 1, type 2, and type 17 cytokines released by liver cells of untreated and uninfected naïve mice was remarkable. Liver cytokines of 23 days-infected mice differed from naïve mice in significant decrease in the levels of IL-33, reported to be dispensable for *S. mansoni* maturation [28]. The most striking finding ($P < 0.01$) involved increase in IL-17, likely responsible for the changes in cellular infiltration in the liver as early as 4 wks PI, compared to naïve mice, described by Costain et al. [29]. Despite the dramatic increase reported by Costain et al. [29] in numbers of hepatic eosinophils

**Table 3. Liver homogenates reactive oxygen species content.**

| | Fluorescence counts of mouse groups* | | | | | |
|---|---|---|---|---|---|---|
| | 1 | 2 | 3 | 4 | 6 | 6 |
| **Liver proteins μg/well** | | | | | | |
| 25 | | | | | | |
| Mean | 25449 | 18407 | 24853 | 31933 | 51791 | 41783 |
| SD | 6104 | 5194 | 14691 | 7323 | 16867 | 11946 |
| Median | 23895 | 17796 | 18549 | 30653 | 52307 | 48181 |
| P vs naïve | | **0.0262** | NS | | | |
| P vs infected | | | NS | **0.0008** | **0.0001** | **0.0002** |
| 50 | | | | | | |
| Mean | 46161 | 35332 | 46734 | 51221 | 80541 | 72897 |
| SD | 16333 | 6800 | 21031 | 9768 | 35823 | 14092 |
| Median | 46076 | 35435 | 37019 | 53307 | 71540 | 75437 |
| P vs naïve | | NS | NS | | | |
| P vs infected | | | NS | **0.0027** | **0.0035** | **<0.0001** |
| 100 | | | | | | |
| Mean | 95466 | 71034 | 89171 | 96528 | 147681 | 129235 |
| SD | 30248 | 6827 | 26669 | 20893 | 37702 | 12526 |
| Median | 94306 | 70362 | 77805 | 89822 | 162754 | 131255 |
| P vs naïve | | 0.0428 | NS | | | |
| P vs infected | | | NS | **0.0055** | **<0.0001** | **<0.0001** |
| 200 | | | | | | |
| Mean | 148788 | 123366 | 148627 | 152656 | 208272 | 199141 |
| SD | 41494 | 9708 | 45547 | 13425 | 73098 | 27919 |
| Median | 145233 | 122128 | 127418 | 149922 | 182390 | 189286 |
| P vs naïve | | NS | NS | | | |
| P vs infected | | | NS | **0.0002** | **0.0057** | **<0.0001** |

*Reactive oxygen species fluorescence counts of liver Triton X-100 extracts of 5–8 mice per group: 1, naïve; 2, infected; 3, alum/infected; and 4, MAP-2 + alum; 5, MAP-1 + MAP-2; and 6, MAP-1 + MAP-2+ alum-immunized mice. P values indicate levels of statistical (ANOVA and Mann-Whitney) differences between mouse groups. NS = not significant.

and macrophages in 4 wks-infected mice, no changes in ARA or ROS hepatic content was recorded in 23 days-infected mice, compared to naïve controls. The adjuvant used, alum (aluminum hydroxide, 130 μg/injection) failed to impact challenge worms' survival or fecundity, associated with induction of significant ($P < 0.05$ to $P < 0.005$) reduction in liver cytokines, including IL-17, ARA, and ROS content when compared to levels recorded in naïve or unimmunized infected mice. It is important to recall that adjuvants, including alum, trigger stromal cells responses at the site of injection [12] and may not impact remote sites.

Compared to naïve and untreated or alum-administered infected mice, MAP-immunized mice showed at 23 days PI, significant ($P < 0.05$ - $<0.0001$) increase in content of hepatic IL-1, ARA, and ROS. Interleukin-1β is predominantly produced by myeloid cells but also by B and T lymphocytes [30–33], and is a key mediator in inflammation initiation, maintenance, and amplification [30–36]. We surmise that elevated IL-1 release induces accumulation and activation of neutrophils, eosinophils, and macrophages. Activation of these inflammatory cells promotes release of free ARA from their cell membranes [37,38] that in turn enhances their ROS production [37,39]. Both ARA [for review see 40] and ROS [41–43] activate schistosomes and cells surface membrane-associated neutral sphingomyelinase (nSMase)-2. Hepatic cells are not

Peptides shared with gut cysteine peptidases

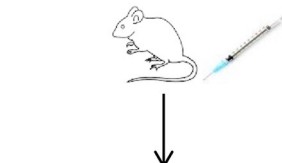

At 23 days PI: the schistosome liver stage

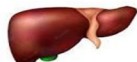

Release and antigen presentation of gut cathepsins followed by interaction with the MAP-specific memory T and B cells

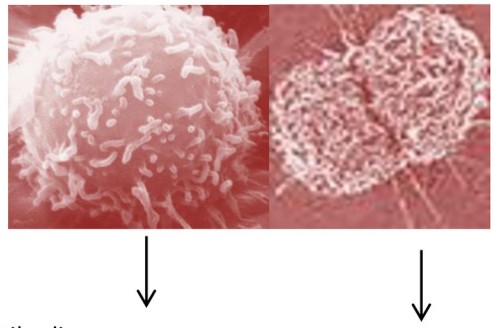

Release of anti-MAP antibodies

That complex with gut cathepsins

And bind to and activate inflammatory cells          Increase in the pro-inflammatory IL-1 in liver

Elevates ARA and ROS content          Activates lipid and cholesterol metabolism

Dead Worms          Fertile Surviving Worms

Percent reduction: 60%          Egg burden: equal or higher than controls

**Fig 8. Proposed mechanisms underlying cysteine peptidases-based vaccine impact on challenge *Schistosoma mansoni*. Icons modified from https://commons.wikimedia.org.**

  

damaged because of lack or very low levels of surface membrane-associated nSMase-2 [42], while nSMase-mediated hydrolysis of worms surface membrane sphingomyelin is a direct killing hit [9,40]. Increase in hepatic cells ROS levels is counteracted by the persistent high content of the anti-oxidant uric acid [44]. Conversely, the juvenile worms surface membrane lipids and proteins are irreversibly oxidized, further impairing the integrity of their outer membrane shield [45–47]. Thus, IL-1-related increase in hepatic free ARA and ROS content in MAP-immunized mice provides an explanation for the recorded highly significant ($P < 0.005$) reduction in challenge worm burden. However, IL-1β promotes triglycerides and cholesterol accumulation in murine and human liver cells [48–50]. Elevated levels of triglycerides, cholesterol, and ARA [48–52] promote the reproductive activities and egg production of the surviving worms, explaining the difficulties in controlling worm egg output in mice immunized with cysteine peptidase in full length [1–10] or peptide [11] constructs (Fig 8).

The solution would be to devise peptide-based formulations that induce increase in IL-1, ARA, and ROS targeting both the lung and liver schistosome stages. Lung-stage larvae are exceedingly sensitive to ARA [40] and ROS [45–47] schistosomicidal effects, while it is premature for IL-1-related activation of lipids metabolism to impact the surviving larvae reproductive functions. Support is provided by the observation that the highest reduction in challenge *S. mansoni* worm burden (76.5%, $P = 0.0006$), and liver and small intestine egg loads (61.6%, $P = 0.0006$; and 57.1%, $P = 0.0023$, respectively) was achieved by outbred mouse immunization with SmCB1 + SmCL3 combined with SG3PDH [5], *S. mansoni* glyceraldehyde phosphate dehydrogenase, a prominent lung-stage larvae excretory-secretory product [1,2,5].

## Conclusions

Peptides common to several gut cysteine peptidases in MAP construct, formulated in mixture and/or combined with the adjuvant, alum elicited highly significant ($P < 0.005$) reduction in challenge worm burden. Memory responses to the immunogens are expected to be expressed at the post-lung stage, notably in the liver. The impairment in challenge worm survival in immunized mice was associated with hepatic increase in the levels of the pro-inflammatory IL-1, and the schistosomicidal ARA and ROS. Interleukin-1-related activation of hepatic lipids, ARA, and cholesterol synthesis likely promotes the reproductive functions and egg formation in the surviving liver-stage schistosomes, explaining the difficulty in reducing parasite egg outputs in mice immunized with cysteine peptidases in full-length or peptidic constructs (Fig 8). Reduction of worm egg output is proposed to be achieved via peptide formulations eliciting memory immune responses targeting both the lung- and liver- schistosome developmental stages.

## Supporting information

**S1 Fig. Parasitological parameters.** Liver granulomas number (A) and diameter (B) were evaluated 7 wks post challenge infection. Statistical (Mann-Whitney) differences between immunized and infected control mice are not significant (NS).
(TIF)

**S2 Fig. Uric acid content.** (A) Columns represent μg uric acid/mg liver proteins in 5–8 individual mice/group. Liver Triton X-100 extracts were assayed for uric acid content in duplicate 50 μg protein samples per well using in parallel two separate Uric Acid Assay Kits. Similar results were obtained and were, therefore, pooled. No significant differences between groups were recorded as assessed by ANOVA. (B) Columns represent μg uric acid/mg wet liver in 5–8 individual mice per groups. Results of uric acid content (μg per mg wet liver) with assays of 2.5

  

and 5 μL/well were similar, and, therefore, pooled. No significant differences between naïve, infected, and MAP-immunized groups were recorded as assessed by ANOVA.
(TIF)

**S3 Fig. Liver arachidonic immunochemistry.** Liver sections of each of 3 naïve (A); infected (B); alum/infected (C); and MAP-2 + alum (D), MAP-1 + MAP-2 (E), and MAP-1 + MAP-2 + alum- (F) immunized mice were reacted with 0.5 μg horseradish peroxidase-linked polyclonal antibody to ARA (MyBioSource, MBS2051576) overnight at 10˚C. The reaction was visualized with Dako Liquid DAB + Substrate Chromogen System. Figures shown are representative of the consistently recorded reactivity for each mouse group on day 23 post *S. mansoni* infection. x100.
(TIF)

## Acknowledgments

Thanks are due to Mr. Abdel Badih Foda for invaluable technical help and animal care.

## Author Contributions

**Conceptualization:** Hatem Tallima, Rashika El Ridi.

**Data curation:** Rashika El Ridi.

**Formal analysis:** Hatem Tallima.

**Funding acquisition:** Hatem Tallima, Rashika El Ridi.

**Investigation:** Hatem Tallima, Rashika El Ridi.

**Methodology:** Hatem Tallima, Rashika El Ridi.

**Writing – original draft:** Hatem Tallima, Rashika El Ridi.

**Writing – review & editing:** Hatem Tallima, Rashika El Ridi.

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
