## [Decision Letter · Decision Letter 0]

5 Dec 2022

Dear Dr El Ridi,

Thank you very much for submitting your manuscript "Increased hepatic interleukin-1, arachidonic acid, and reactive oxygen species mediate the protective potential of peptides shared by gut cysteine peptidases against Schistosoma mansoni infection in mice" for consideration at PLOS Neglected Tropical Diseases. As with all papers reviewed by the journal, your manuscript was reviewed by members of the editorial board and by several independent reviewers. In light of the reviews (below this email), we would like to invite the resubmission of a significantly-revised version that takes into account the reviewers' comments. 

All the reviewers agree that while a potentially interesting study, in its current form the manuscript is difficult to read. Substantial editing is required particularly regarding how the data is presented, before the manuscript would be suitable for publication in PLoS NTD. In this instance, the authors can decide to make the substantial revisions required, addressing the comments raised by the reviewers and continue with this review process or if substantial time is required for the editing process, I recommend that the authors withdraw this submission and re-submit the manuscript as a new submission.

We cannot make any decision about publication until we have seen the revised manuscript and your response to the reviewers' comments. Your revised manuscript is also likely to be sent to reviewers for further evaluation.

Sincerely,

Krystyna Cwiklinski, PhD

Academic Editor

Cinzia Cantacessi

Section Editor

All the reviewers agree that while a potentially interesting study, in its current form the manuscript is difficult to read. Substantial editing is required particularly regarding how the data is presented, before the manuscript would be suitable for publication in PLoS NTD. In this instance, the authors can decide to make the substantial revisions required, addressing the comments raised by the reviewers and continue with this review process or if substantial time is required for the editing process, I recommend that the authors withdraw this submission and re-submit the manuscript as a new submission.

Reviewer's Responses to Questions

**Key Review Criteria Required for Acceptance?**

**Methods**

-Are the objectives of the study clearly articulated with a clear testable hypothesis stated?

-Is the study design appropriate to address the stated objectives?

-Is the population clearly described and appropriate for the hypothesis being tested?

-Is the sample size sufficient to ensure adequate power to address the hypothesis being tested?

-Were correct statistical analysis used to support conclusions?

-Are there concerns about ethical or regulatory requirements being met?

Reviewer #1: (No Response)

Reviewer #2: The two authors of this story present a study in a mouse model that aims to map a highly relevant clinical problem. A vaccine against schistosomes is one of the most important vaccines to be developed, and very difficult due to the complexity of the pathogen and its developmental stages. The vaccine combination MAP-2 in combination with alum and/ or MAP-1, resulted in a 60% worm burden reduction of a challenge infection, which was explained by the authors with significantly increased levels of IL-1 ARA and ROS in the liver.

The objectives of the study are clearly stated. However, why an older formula had to be modified is just stated but not justified. It would be nice to hear about expected advantages. 

The study design, including sample sizes, is appropriate to measure the success of the vaccine combination on a challenge infection.

Reviewer #3: The Methods are complete but could be written much more succinctly since these have been reported in several previous manuscripts by the authors

**Results**

-Does the analysis presented match the analysis plan?

-Are the results clearly and completely presented?

-Are the figures (Tables, Images) of sufficient quality for clarity?

Reviewer #1: (No Response)

Reviewer #2: The results are somewhat disorganized, the graphs are formatted differently, and some of the X-axis labels are missing. Different colors and fonts reduce the clear representation. It appears as if a prefinal version has been submitted.

Reviewer #3: The data is intersting but presented very poorly in graphs, figures and Tables that are not easy to follow. Much of this data should be placed in the Supplementary data. The text in this section should be more focused on the more relavnt results.

**Conclusions**

-Are the conclusions supported by the data presented?

-Are the limitations of analysis clearly described?

-Do the authors discuss how these data can be helpful to advance our understanding of the topic under study?

-Is public health relevance addressed?

Reviewer #1: (No Response)

Reviewer #2: The conclusion of the abstract "A major mechanism of schistosomiasis vaccine protective potential is now clarified." is not supported by the data. The effect of reduction of worm burden is associated with .... or maybe due to...., elevation of expression of three genes in the liver does not represent a major mechanism.

Limitations and public health relevance are not described.

Reviewer #3: The Discussion followed by Conclusion is again very voluminous and could be reduced significantly

**Editorial and Data Presentation Modifications?**

Reviewer #1: (No Response)

Reviewer #2: Given the high relevance of the study, vaccines are urgently needed, and given the expert status of the authors, this study disappoints because of the poor quality of expression, language, and presentation. It appears that a prefinal version was submitted. The manuscript urgently needs revision, preferably with the assistance of a native speaker. The abstract needs to be revised since important results are missing. 

The following are just a few examples of typos. The expression needs to be revised as well:

-L77: Revision of the phrase is necessary

-L138: granuloma numbers

-L149: Are the following tests performed with fresh liver tissue?

-L155: better reaction tubes

-L171: Arachidonic acid ...assay?

-L172: Free ARA abbreviation and later L180 Arachidonic acid

-L217: Which results justify the conclusion of transit to exit points?

-L241: This line is empty. Are there no values? 

-Fig 1: Labeling of x-axis

-L 267 and 268: Revision of the phrase is necessary.

-L306: egg granulomas

-Fig 3: The illustrations are completely inadequate to corroborate the statement in L272.

-L319: correction of the position of the labelling

-L358 Il-17 ) or } and Protein or proteins

-Fig.6: Nothing is seen on the pictures.

-L435: Who recommended this?

-L457: Increase

-L457: Please explain FhCL1

Reviewer #3: (No Response)

**Summary and General Comments**

Reviewer #1: This manuscript presents an investigation into the protective potential of a vaccine comprised of multiple antigen peptide constructs designed based on the sequences of parasite cathepsins and delivered with and without alum as an adjuvant. The results presented suggest these vaccine candidate resulted in a significant reduction in worm burden following a challenge infection with S. mansoni in outbred mice. The combination of MAP-1, MAP-2 and alum led to the most significant reduction in worm burden and the smallest size of intestinal granuloma. Some measures of immune response (cytokines, ROS) were measured in both liver and intestine of all animals.

The protection achieved in the vaccinated animals is remarkable and builds on the previous reports from this group regarding the efficacy of cysteine peptidases in anti-helminth vaccines. 

I would suggest that the data could be presented differently. I am not sure that Fig 1 is necessary as this data would be better placed within Table 1. Similarly, the data presented in Fig 2 should be incorporated into Table 2.

The authors then performed a number of immunological assays to correlate with the protective effect of the vaccines. This section of the manuscript was very difficult to read as the data is presented in a variety of different formats.

• Figure 4E – I would suggest that the treatment names are added to the lower axis of the graph rather than as a separate line of text – it is difficult to read as it is currently presented.

• Fig 5 – each of these graphs should be presented as a histogram with the treatment groups added to the lower axis – similar to the current presentation of Fig 4A. 

• How many mice are presented in Fig 5? This information needs to be added to the legend

• Although the statistical significance of data is mentioned in the text for Figure 5, there is no addition of significance markers to the graphs – can these please be added.

In the descriptive text for the cytokine analysis there is multiple mentions of correlation between worm burden and cytokines – however, I would argue that these statements do not hold as if this was indeed the case there should be true correlation across all the treatment groups. For example there is a claim that the increase in liver IL-1 correlated with the reduced worm burden – but if this was the case, then the MAP-1+ MAP-2 +Alum, which had the lowest worm burden, should have the highest IL-1, but the levels in this group look very similar to the infected cohort.

• There are also a number of mechanistic conclusions drawn in the text (for example: led to modification of the host innate and acquired immune responses) without appropriate analysis being performed of relevant data presented.

• I would suggest that the authors significantly temper the language and conclusions they are drawing from this data.

Figure 6A – there is no need for both a table and a graph here – one or other will suffice. The graph would be a preferred option, although the axis labels needs to include the treatment groups and the associated statistical significance markings need to be added to the graph

Figure 6B is quite unclear and perhaps redundant given the quantified assay presented in Figure 6A?

The discovery that the vaccinated animals had significantly increased Arachadonic acid compared to the infected controls is interesting; however these animals also had significantly increased levels of ROS in their livers which appears slightly contradictory – although this is partly addressed in the discussion of the paper.

However, the discussion is a little hypothetical and a large portion of the theories presented seem to depend on the levels of IL-1 found within the liver – which as I mentioned previously seems to be variable affected by the vaccine preparations and is not a clear correlate with protection, or with the increased levels of ARA and ROS

The authors claim to have uncovered the immune mechanisms that mediate the protective effect (i.e IL-1, ROS and ARA) – however to support these claims some more direct mechanistic analysis would be required – again, I would suggest the authors apply a more measured approach to their discussion and conclusions.

The writing needs to be edited as there are a number of issues with the English, and sentence structure, throughout making parts of the text difficult to understand.

Reviewer #2: Given the high relevance of the study, vaccines are urgently needed, and given the expert status of the authors, this study disappoints because of the poor quality of expression, language, and presentation. It appears that a prefinal version was submitted.

Reviewer #3: The paper represents one in a long series of similar reports by this team purpsung cysteine proteases as vaccines against schistosomes, all are difficult to follow. While I agreed with the studies and outcomes these papers have seemed very similar and no taking big enough steps to warrant publication as a major paper. The authors should keep a focus and report only the most important data instead of rambling on about mechanisms of protection that are not fully elucidated here - the perfect cytokine balance is not know. The figures are poorly prepared and presented - are they all needed? Could they just not be added to supplementary section or simple referred to data not shown. 

The paper isn't making a step-change and in my opinion is best suited to a vaccine-related journal.

Some questions -

1. Why exactly were these short peptide chosen. Do they also match with host cysteine proteas and have the potential to induce auto-immune responses?

2. Why was MAP chosen rather than linking the peptides to another carrier? Would they work with another carrier?

3. Wasn't sure of the meaning in 74 - 76 sentence - are they authors saying that the antibodies to peptides do not block the enzymes activity. If not how do they block the enzyme function.

4. line 110 - 39 rather than 40?

5. Line 125 - not sure why egg response would interfere with peptide cytokine responses this statement is not clear.

PLOS authors have the option to publish the peer review history of their article (what does this mean?). If published, this will include your full peer review and any attached files.

Reviewer #1: No

Reviewer #2: No

Reviewer #3: No
---

## [Decision Letter · Decision Letter 1]

30 Jan 2023

Dear Dr El Ridi,

Thank you very much for submitting your manuscript "Increased hepatic interleukin-1, arachidonic acid, and reactive oxygen species mediate the protective potential of peptides shared by gut cysteine peptidases against Schistosoma mansoni infection in mice" for consideration at PLOS Neglected Tropical Diseases. As with all papers reviewed by the journal, your manuscript was reviewed by members of the editorial board and by several independent reviewers. The reviewers appreciated the attention to an important topic. Based on the reviews, we are likely to accept this manuscript for publication, providing that you modify the manuscript according to the review recommendations. 

Although the authors have addressed the comments raised by the first review, further revision of the manuscript is required before being suitable for publication particularly relating to the results section - see reviewers comments which need to be addressed.

Sincerely,

Krystyna Cwiklinski, PhD

Academic Editor

Cinzia Cantacessi

Section Editor

Although the authors have addressed the comments raised by the first review, further revision of the manuscript is required before being suitable for publication particularly relating to the results section - see reviewers comments which need to be addressed.

Reviewer's Responses to Questions

**Key Review Criteria Required for Acceptance?**

**Methods**

-Are the objectives of the study clearly articulated with a clear testable hypothesis stated?

-Is the study design appropriate to address the stated objectives?

-Is the population clearly described and appropriate for the hypothesis being tested?

-Is the sample size sufficient to ensure adequate power to address the hypothesis being tested?

-Were correct statistical analysis used to support conclusions?

-Are there concerns about ethical or regulatory requirements being met?

Reviewer #1: (No Response)

Reviewer #3: Yes,

**Results**

-Does the analysis presented match the analysis plan?

-Are the results clearly and completely presented?

-Are the figures (Tables, Images) of sufficient quality for clarity?

Reviewer #1: (No Response)

Reviewer #3: Yes

**Conclusions**

-Are the conclusions supported by the data presented?

-Are the limitations of analysis clearly described?

-Do the authors discuss how these data can be helpful to advance our understanding of the topic under study?

-Is public health relevance addressed?

Reviewer #1: (No Response)

Reviewer #3: Yes, I'm happy with the authors' interpretation

**Editorial and Data Presentation Modifications?**

Reviewer #1: (No Response)

Reviewer #3: (No Response)

**Summary and General Comments**

Reviewer #1: This revised version of the manuscript is much improved with the data now presented in more consistent and understandable form. The text has also been positively edited throughout. 

However, in the results section describing the measure of cytokines in tissue (around Line 305) a number of statements which claim correlations between specific cytokines and a protective effect remain. These are hypothetical and as such belong in the discussion (where this is addressed) and should only be in the results section of actual statistical correlations have been performed. This section of next needs to be edited accordingly such that only the actual results are described here and not a proposal as to the function/outcome of these changes.

There remains some minor issues with the English in parts (although improved greatly since the original submission) - I would suggest a native english speaker performs an edit/read through prior to resubmission.

Reviewer #3: I'm happy with the responses

PLOS authors have the option to publish the peer review history of their article (what does this mean?). If published, this will include your full peer review and any attached files.

Reviewer #1: No

Reviewer #3: No

Figure Files:

Data Requirements:

Reproducibility:

References

---

## [Editor Report · Decision Letter 2]

13 Feb 2023

Dear Dr El Ridi,

We are pleased to inform you that your manuscript 'Increased hepatic interleukin-1, arachidonic acid, and reactive oxygen species mediate the protective potential of peptides shared by gut cysteine peptidases against Schistosoma mansoni infection in mice' has been provisionally accepted for publication in PLOS Neglected Tropical Diseases.

Best regards,

Krystyna Cwiklinski, PhD

Academic Editor

Cinzia Cantacessi

Section Editor

The authors have now addressed the comments raised by the review process and the manuscript is suitable for publication in PLOS NTD.

---

## [Editor Report · Acceptance letter]

6 Mar 2023

Dear Dr El Ridi,

We are delighted to inform you that your manuscript, "Increased hepatic interleukin-1, arachidonic acid, and reactive oxygen species mediate the protective potential of peptides shared by gut cysteine peptidases against Schistosoma mansoni infection in mice," has been formally accepted for publication in PLOS Neglected Tropical Diseases.

Best regards,

Shaden Kamhawi

co-Editor-in-Chief

Paul Brindley

co-Editor-in-Chief
